# Efficacy and Safety of Biological Therapies and JAK Inhibitors in Older Patients with Inflammatory Bowel Disease

**DOI:** 10.3390/cells12131722

**Published:** 2023-06-26

**Authors:** Walter Fries, Giorgio Basile, Federica Bellone, Giuseppe Costantino, Anna Viola

**Affiliations:** 1Gastroenterology, Department of Clinical and Experimental Medicine, University of Messina, 98125 Messina, Italy; giuseppe.costantino@polime.it (G.C.); anna.viola@unime.it (A.V.); 2Unit of Geriatrics, Department of Biomedical and Dental Science and Morphofunctional Imaging, University of Messina, 98125 Messina, Italy; giorgio.basile@unime.it; 3Unit of Internal Medicine, Department of Clinical and Experimental Medicine, University of Messina, 98125 Messina, Italy; federica.bellone@unime.it

**Keywords:** frailty, comorbidities, Crohn’s disease, ulcerative colitis, infections, malignancies

## Abstract

With the introduction of more and more monoclonal antibodies selectively targeting various mediators of the immune system, together with Janus-Kinase (JAK)-inhibitors with variable affinities towards different JAK subtypes, the available therapeutic options for the treatment of inflammatory bowel diseases (IBD) have undergone an acceleration in the last five years. On the other hand, the prevalence of IBD patients over 65-years-old is steadily increasing, and, with this, there is a large population of patients that presents more comorbidities, polypharmacy, and, more frequently, frailty compared to younger patients, exposing them to potentially major risks for adverse events deriving from newer therapies, e.g., infections, cardiovascular risks, and malignancies. Unfortunately, pivotal trials for the commercialization of new therapies rarely include older IBD patients, and those with serious comorbidities are virtually excluded. In the present review, we focus on existing literature from pivotal trials and real-world studies, analyzing data on efficacy/effectiveness and safety of newer therapies in older IBD patients with special emphasis on comorbidities and frailty, two distinct but intercorrelated aspects of the older population since age by itself seems to be of minor importance.

## 1. Introduction

Approval of new therapies by the Food and Drug Administration (FDA) and/or European Medicines Agency (EMA) is based on pivotal trials showing efficacy and safety of new therapeutic principles in patients with inflammatory bowel diseases (IBD). The population of these pivotal phase III studies does not represent the whole patient population because of strict inclusion/exclusion criteria [1]. Not only age but comorbidities, such as liver, kidney, or cardiovascular disease, together with previous diagnoses of dysplasia or malignancy, represent the major reasons for excluding a patient from randomized clinical trials (RCT) [2]. These limitations raise the need for post-marketing surveillance and real-world data to confirm the effectiveness of new therapies in elderly patients and to recognize hidden problems not identified in phase III trials. One of the best-known examples is represented by the tuberculosis (TB) risk with anti-TNF agents, having been identified only after approval of this inhibitor by the regulatory authorities [3] and the subsequent recommendations for an appropriate screening and preventive therapy. Latent TB is more frequent in the elderly population [4], i.e., a population not included in the pivotal anti-TNF trials.

## 2. Age, Comorbidities, and Frailty

Aging is not only a chronologic process but also a complex, dynamic process, resulting from interaction between genetic and environmental factors. It is additionally characterized by the progressive loss of functional reserves, which negatively affects individual adaptation to contextual demands, including a potential increase in adverse drug reactions (ADR).

In recent decades, the number of people with chronic disorders, including IBD, has dramatically increased worldwide. In a Canadian study, older patients with IBD, including elderly-onset IBD patients (diagnosed at an age ≥65 years), and those transitioning to older age with longstanding diseases with a yearly prevalence increase of nearly 3%, are calculated to reach a total prevalence of 1% by the year 2030 [5]. Similar numbers are reported from Finland, where 33% of prevalent cases were reported to be in the age group >60 years [6].

With advancing age, the number of comorbidities frequently increases. The Charlson Comorbidity Index [7] (CCI) and the Elixhauser index [8] are weighted indices considering the number and severity of comorbidities, both developed to predict mortality in the general population and are frequently used in descriptive data concerning the older IBD population [9]. Where investigated, increasing values for the CCI and/or Elixhauser index were found to be associated with an increase in worse outcomes with various biological therapies [10,11,12].

A typical feature of advanced age, often in the presence of multimorbidity, is frailty. Frailty is a highly prevalent condition with advancing age and is characterized by an increased vulnerability to stressors due to reduced homeostatic reserves [13]. In fact, older adults with IBD are characterized by a multifactorial clinical picture in which age-related physical, functional, and psychological symptoms may coexist [14,15]. Despite the acknowledgement of the importance of frailty, until a few years ago, no study assessed the cognitive or social status or the functional performance in older adults with IBD [16]. For much time, chronological age was the only objective measure for the assessment of the effectiveness and safety of biological therapies in elderly IBD patients. Frailty is more prevalent in IBD compared with matched non-IBD subjects, reaching 12% of IBD patients [17]. Although there are several tools to assess frailty, two models are commonly used: The Fried frailty phenotype [18] and the deficit accumulation model of Rockwood [19]. The first model of frailty is based on the presence of at least three out of five criteria investigating specific physical variables (weight loss, fatigue, reduced gait speed, poor handgrip strength, and sedentary habits), with the central pathophysiological element being sarcopenia, the later by itself may negatively affect clinical outcomes in IBD [20]. Conversely, the Rockwood model conceives a multidimensional frailty status as the result of the accumulation of deficits. Accordingly, frailty is measured by the Frailty Index (FI), which is defined as the ratio between deficits present in an individual and the total amount of age-related health variables considered. A very recent cohort study showed that in elderly outpatients with IBD, the presence of deficits (≥2 domains out of 5 considered: Somatic, functional activities of daily living, physical capacity, mental and social status) was associated with IBD disease activity, and with a higher disease burden [16,21]. More recent scores like the frailty risk score [22] and variants [23] have been used in IBD patients, showing a critical prevalence in IBD patients and association with adverse outcomes, such as morbidity, hospital admissions, and readmissions [24], increased risk of infections, especially in the presence of concomitant immunosuppressive treatments [25], and mortality [17,26].

Therefore, evaluation of frailty, possibly with simplified assessment tools, is essential in patients with IBD for a better prognostic definition and to guide therapeutic choices or to adapt, where possible, dosing regimens to minimize adverse events.

## 3. Biological Therapies

Currently, available biological therapies include monoclonal antibodies such as anti-TNF agents (infliximab (IFX), adalimumab (ADA), golimumab (GOL)), anti-α4β7 integrins (Vedolizumab, (VEDO)) and an antibody against the p40 subunit shared by interleukin (IL)-12 and IL-23, (Ustekinumab, (USTE)). With the exception of GOL, licensed only for ulcerative colitis (UC), all others are licensed for both Crohn’s disease (CD) and UC. While drug-drug interactions for monoclonal antibodies have never been reported, an important aspect is their long half-life compared to other therapies. The sequence of their use, i.e., which drug to use as first-line therapy or in a later treatment line and the degree of immunosuppression that they exert, are important variables to consider with their use in older IBD patients.

## 4. Anti-Tumor Necrosis Factor (TNF) Agents

Anti-TNF therapies have precise contraindications, such as cardiac insufficiency (grade 3 or 4), active TB, or severe concomitant infections [27]. Renal insufficiency does not imply dose reductions. Circulating through levels of anti-TNFs correlate with clinical efficacy [28] and lengthening of dose intervals to reduce drug exposure to mitigate potential side effects is currently under investigation in non-elderly patients [29].

Although the first anti-TNFs (IFX and ADA) were approved 20 years ago, only a minority of studies have addressed their effectiveness in elderly patients, whereas a more consistent body of evidence is available concerning safety. No real-world study has ever investigated GOL in older patients with UC.

**Effectiveness**: From the first retrospective studies on the effectiveness of anti-TNFs as first-line therapy carried out in numerically very limited studies, equal [30] or reduced response [31] or higher withdrawal rates during therapy [32,33] were reported in mixed biologic-naïve older IBD populations compared with younger patients. Withdrawal was mostly due to adverse events, namely infections, whereas loss of response was similar compared to younger patients (Figure 1). 

Age or comorbidities were found to be associated with the discontinuation of therapy and with ADRs. In an analysis of pooled data from RCTs with IFX and GOL, no difference was found between older and younger patients in terms of achieving and maintaining remission with these anti-TNFs used as first- or second-line therapies [34].

In a propensity score-matched comparison study with VEDO, IFX showed a lower probability of treatment failures in older patients with comparable CCI and treatment-line in both groups [35]. 

**Figure 1 cells-12-01722-f001:**
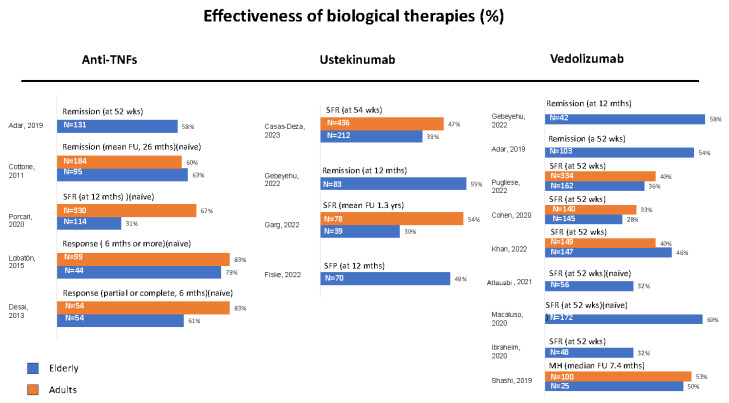
Real-world studies on the effectiveness of biological therapies in elderly and adult IBD patients. The most representative endpoint of each study was selected. Data are expressed as percentages according to different outcomes. SFR = steroid-free remission, SFP = steroid-free persistence, FU = follow-up, MH = mucosal healing, naïve: Patients without prior biologic treatment, studies including a younger control group of patients treated with the same biologic drug are represented with parallel bars, studies comparing different therapeutic drugs are allocated to the respective column, Ref [36] was excluded because only treatment failure was calculated [30,31,32,33,37,38,39,40,41,42,43,44,45,46,47,48].

**Safety:** In the first report by Cottone et al. [30], more severe infections and an increased mortality in older IBD patients treated with anti-TNFs (IFX or ADA) compared with other non-biological treatments were reported. An increased risk of severe adverse events was reported in a pooled analysis of RCTs in elderly UC patients but without a sure attribution to anti-TNFs (GOL and IFX) [34].

A four-fold increased risk of opportunistic infections with IFX monotherapy was reported in patients >50 years of age, together with a further increase in the case of combination therapies [36]. A more recent systematic review with meta-analysis on biologics (IFX, ADA, VEDO) concluded that these therapies were likely to increase the risk of opportunistic and serious infections in older (>60 years) IBD patients [49]. A study on a large database, however, showed that anti-TNFs, together with systemic steroids, increased the risk of pneumonia and pneumonia-related hospitalizations [50].

In comparison with other biological treatments, e.g., VEDO, IFX showed no major risk of infections in older patients with fewer comorbidities [35] (p. 6) but a significantly higher risk of infection-related hospitalizations in older patients with more severe comorbidities [51]. Similar data were reported when comparing anti-TNFs with VEDO or USTE, showing no overall difference in older patients but more infection-related hospitalizations in the anti-TNF-treated patients, with a CCI of >1 [52].

Interestingly, in a large administrative claims-based cohort study, anti-TNF treatment showed lower mortality in older CD with at least two major comorbidities, together with a lower risk of major cardiovascular events (MACE) and risk of hip fractures compared with prolonged steroid use [53]. For older UC patients, the risk of mortality and serious infections was similar in steroid or anti-TNF-treated patients. This finding was reproduced later, confirming a reduced mortality with anti-TNF compared with steroids in older patients both with CD and UC [54].

Of interest, frail IBD patients who respond to anti-TNF therapy may experience an improvement in frailty over time [55].

Concerning cancer risk, analyzing the data of a very large US database, Khan et al. showed an increased risk for solid organ malignancies (colorectal, prostate, lung, urinary tract, and female breast cancer) and non-Hodgkin lymphomas in elderly IBD patients, independent from medications [56].

Variables associated with that increased risk were male sex, disease duration, CCI of at least 1 point, polypharmacy, solid organ transplant, hypertension, and chronic obstructive pulmonary disease. A potential increased cancer risk induced by the systemically acting anti-TNFs was investigated in several studies, but no clear association of anti-TNFs was reported in a prospective study with prolonged follow-up [57] or in a systematic review with meta-analysis with different biologics (IFX, ADA, VEDO) [49].

In summary, anti-TNFs appear to have similar effectiveness in older patients with IBD compared with younger patients but bear an increased risk of infections (opportunistic or serious). Compared with prolonged steroid use, their safety profile would appear to be superior. Use in an elderly/frail IBD population needs careful patient profiling, whereas use in elderly/non-frail patients does not seem to be of particular risk. A complete vaccination panel to reduce potential infectious risk where possible is mandatory. The increased cancer risk in the elderly IBD population by itself and in association with anti-TNFs warrants an appropriate surveillance protocol.

## 5. Anti–α4β7 Integrin Vedolizumab

The action of VEDO, different from anti-TNFs, is limited to the α4β7 integrins present in the gastrointestinal tract, thus, VEDO does not seem to have a systemic effect and, therefore, may be safer in older IBD patients. 

**Effectiveness**: Most studies investigating the efficacy of VEDO in the elderly IBD population were very small. Thus, a retrospective case-control study showed no difference in effectiveness in older CD patients compared with a younger mixed IBD cohort [37], and similar results came from a retrospective UK study on a mixed IBD population [38]. Subsequently, a good clinical response was reported in older biologic-naïve CD and UC patients without comparator [39] and in older patients with contraindications to anti-TNFs [40]. None of these four studies assessed comorbidities or polypharmacy (Figure 1).

In a large study carried out on US veterans comparing effectiveness in older IBD patients with a significantly higher CCI score with effectiveness in a younger IBD population, similar outcomes were reported in both populations [41].

Comorbidities were considered instead in a retrospective study comparing a mixed, mostly anti-TNF experienced patient population >60 years of age against patients <40 years without any evidence of differences in effectiveness [42] and, in a partially prospective study, older patients with UC showed lower persistence compared to matched younger controls, whereas in CD no differences were observed between the two groups in terms of effectiveness [43]. In this study, a CCI of >2 in CD patients was positively associated with lower persistence. In a large study comparing VEDO with anti-TNFs in a mixed, partially anti-TNF-experienced population, VEDO showed a higher risk of failure in CD but not in UC [35] (p. 6). 

**Safety:** In the following two retrospective studies on safety in older IBD patients, VEDO therapy was analyzed against anti-TNF therapies reporting, in the first, a comparable risk for ADR [44] in the two treatment arms without considering CCI or, in the second, a reduced risk for ADR in the VEDO groups (both, CD and UC) in terms of serious infections and infection-related hospitalizations [58]. In the latter study, in both groups, approximately half of the patients had a CCI of ≥2. In the aforementioned partially retrospective study [43] (p. 8), a CCI score <2 represented a protective factor against ADR. In a comparison study on safety with USTE, in both treatment arms, VEDO and USTE, comorbidities and not age were associated with worse outcomes in terms of infections, but the groups were somewhat different, i.e., there were only CD patients in the group treated with USTE [11].

The most frequent ADRs in all studies on VEDO were upper or lower respiratory tract infections, septicemia, and intestinal infections, including *Clostridioides difficile*. 

Cancer seems not to be increased with VEDO treatment, and older male patients had a similar risk compared to patients taking only mesalazine [58]. 

In summary, VEDO represents a valid alternative to anti-TNFs in the case of significant comorbidities or contraindications to the latter. Importantly, also with VEDO, in the presence of comorbidities, more frequent infections have to be expected, making vaccinations as a preventive measure mandatory.

## 6. Ustekinumab

Ustekinumab (USTE), a monoclonal antibody against the p40 subunit present of interleukin (IL)-12 and IL-23, was first licensed by EMA in 2010 for the treatment of CD and in 2019 for UC.

**Effectiveness**: In the first report on 70 elderly CD with a mean CCI of 4.14, USTE showed a good effectiveness and safety profile with a median treatment of 16 months. More than half of these patients were anti-TNF and/or VEDO experienced [45]. Another study comparing a small cohort of older CD patients compared with younger patients found a lower probability of achieving steroid-free remission in patients ≥65-years-old [46]. No information was given concerning CCI or non-IBD comedications. 

In a propensity score-matched analysis, a similar effectiveness of USTE versus VEDO was reported in older CD patients with a comparable CCI score [47]. Finally, the largest elderly CD cohort with a prolonged follow-up came from the ENEIDA registry and compared elderly with matched younger patients according to prior anti-TNF treatment and smoking habits. No differences were found between elderly and non-elderly patients concerning effectiveness [48]. Of note, the median CCI was 1 in the elderly patient group (Figure 1).

No study is currently available on elderly UC patients treated with USTE.

**Safety:** Generally, the safety profile of USTE seems good and, in the smaller studies, serious infections were associated with concurrent steroid use but not with CCI [46]. In the aforementioned comparison study with USTE/VEDO, a similar rate of infections was reported for the two study drugs, and concomitant steroids were present in 20–25% of patients [47]. In this study, only on univariate analysis was CCI associated with serious infections. In the larger ENEIDA registry study, serious infections were reported in 7.08% of elderly patients and in 7.34% of younger patients [48]. Concerning malignancies, the ENEIDA registry reported malignancies occurring more frequently in comparison with younger patients treated with USTE [48], whereas in comparison with VEDO a similar rate of malignancies was reported [47]. 

In summary, data on USTE are scant and, in terms of effectiveness, no data on UC are available. Similar to VEDO, USTE seems to represent a valid alternative to anti-TNFs. The presence of comorbidities, however, should imply careful monitoring of patients during therapy and, prior to the first USTE prescription, vaccinations as a preventive measure are mandatory.

## 7. Janus Kinase Inhibitors (JAK)-Inhibitors

Janus Kinase (JAK) Inhibitors were introduced for the treatment of rheumatic diseases more than 10 years ago, and several years later, Tofacitinib (TOFA) was the first to be licensed for UC, followed, most recently, by Filgotinib (FIL) and Upadacitinib (UPA), both always for UC. The main differences between these JAK inhibitors lie in their different selectivity for the different JAK subtypes, with FIL and UPA being more selective for Jak-1. Other differences are based on their metabolism and potential for drug-drug interactions (see Table 1), both important factors to keep in mind in older patients. TOFA is metabolized and eliminated mainly by the liver and, to a lesser extent, by the kidney, thus dose adjustments are needed in patients with moderate liver disease or severe kidney disease [59]. FIL is mainly eliminated in the urine, and dosing should be reduced in severe kidney disease, whereas an age of >75 years does not influence its pharmacokinetic properties [60]. Finally, it would appear that UPA does not need any dose adjustment in moderate liver or kidney diseases [61].

Unlike biological monoclonal antibodies, JAK inhibitors are characterized by a rapid onset of action and by a very short half-life (5–6 h), making them potentially easier to manage, especially in the event of infections [59].

Since there is very little data on efficacy and safety in the older UC population, we tried to extrapolate data from rheumatic diseases, keeping in mind that higher dosing regimens were approved for UC, at least during induction of treatment.

**Efficacy:** Concerning efficacy of JAK inhibitors in older UC patients, there is very little data for TOFA and FIL and virtually no data for UPA.

Only one study addressed the efficacy and safety of TOFA in elderly patients [64], analyzing data from the pivotal trials. All patients were treated with an induction regimen of 10 mg BID and, thereafter on maintenance with either 10 mg BID or 5 mg BID. Data on efficacy showed no differences in patients ≥65 years compared to young patients, and no differences were found between different dosing regimens, but the number of patients included was rather small. Likewise, in rheumatologic studies, only modest efficacy differences were found between 5 mg BID, or 10 mg BID [65,66], and the 5 mg BID dosing was finally licensed.

The efficacy of FIL in UC patients between 60- and 75-years-old seems to be comparable with younger patients in a post-hoc analysis of the pivotal trials but, again, with very low numbers [67].

**Safety:** Concerning all JAK inhibitors, a recent EMA warning [68] recommended limiting use in patients >65 years, smokers, and those at risk of malignancies and cardiovascular problems, and to use them only if no other therapeutic option is available.

This warning derived from a post-authorization trial on TOFA versus anti-TNF in patients with rheumatoid arthritis aged >50 years with at least one risk factor for cardiovascular disease showing an increased risk of MACE and cancer in patients treated with TOFA [69,70]. Other studies carried out on large insurance databases, however, did not confirm such excess risks in TOFA-treated patients with rheumatoid arthritis for neither MACE [71] nor cancer [72]. Similarly, in non-elderly patients with UC, except for Herpes (H) zoster, no increased risk for TVP or MACE was reported in UC patients analyzed [73], or the risk was deemed comparable with that of anti-TNFs [74].

Venous thromboembolism (VTE) is well known to occur in IBD [75], especially in the elderly when hospitalized with severe disease [76] and/or when treated with steroids [77]. For this reason, all hospitalized IBD patients should receive prophylactic treatment [78]. No agreement has yet been reached on prophylaxis in outpatients [79].

Another important ADR is infections. Indeed, JAK inhibitors are included in the highest risk group for infections after anti-TNFs [80]. H. zoster reactivation is frequent with all already licensed JAK inhibitors, especially in patients treated with high doses [81,82]. It should be kept in mind that patients with IBD per se are at a higher risk for H. zoster compared with non-IBD controls [83] and that age and immunosuppression (including steroids and biological therapies) increase this risk further [84].

An increase in serum levels of low-density lipoprotein cholesterol and high-density lipoprotein cholesterol, at least with TOFA and UPA, may be observed during the induction period and thereafter may require pharmacological treatment [85].

## 8. Prevention or Mitigation of Adverse Events with JAK Inhibitors

Of course, the easiest approach to lower or eliminate a potential drug-induced danger is to limit or avoid the use of JAK inhibitors in high-risk groups, i.e., elderly patients. However, this would imply that access to a promising group of drugs is foreclosed to a large population of UC patients.

Similar to anti-TNF agents, we have to learn to identify patients with important risk factors and minimize potential adverse events during drug exposure.

Risk assessment for MACE must be carried out in every patient before starting JAK inhibitors. Generic risk assessment tools, e.g., the SCORE2-OP score [86], have been developed for the older population, but no score has ever been validated in UC patients [87]. IBD patients, above all, have an increased cardiovascular risk [88].

Cholesterol levels should be checked prior to therapy, during induction, and at regular intervals thereafter, and, in the case of significantly elevated levels, appropriate measures should be offered to patients.

The risk assessment also includes prevention of thromboembolic events. Similar to rheumatoid arthritis, UC represents per se a risk factor for VTE together with older age [75,76]. Whereas for hospitalized patients, preventive measures with low-molecular heparins are included in all guidelines, no such prevention has yet been considered in patients treated with JAK inhibitors. To date, no biomarkers have yet been identified that may be linked to thromboembolism in patients with rheumatoid arthritis [89] or UC. 

Concerning infections, similar to the correct prevention of infections with biological therapies, all patients should undergo the recommended screening and vaccinations [80]. Importantly, the prevention of H. zoster reactivation with the recombinant vaccine Shingrix^®^ should be offered to all patients. In a recent paper using a Markov model, vaccination with recombinant zoster vaccine was cost-effective in Crohn’s >30 years of age and in UC >40 years [90].

## 9. Conclusions

In conclusion, the choice of therapy in difficult-to-treat older IBD patients, keeping in mind that surgery is still an alternative option in UC, should favor safer treatments with a high probability of achieving the therapeutic target, i.e., clinical remission and, at present, a recommendation on a preferred sequencing cannot be given [91,92].

In a former, excellent review on this topic [93], three scenarios were identified by weighing the following variables: The importance of IBD-related complications, potential treatment-related complications, and frailty, including potential non-IBD chronic disease-related complications. The authors suggest anti-TNFs as a first-line approach in the first category, USTE or VEDO in the second, but no precise indications were given for the third category, i.e., patients with more serious frailty. This latter open question carries the risk of major use of steroids in older IBD patients despite the evidence of its deleterious effects in terms of infections, hip fractures, thromboembolic risk, etc.

Further research is urgently needed to investigate safety better in older IBD patients with concurrent comorbidities and frailty, and the following questions need to be addressed, especially but not only for the use of JAK inhibitors: (a) A comprehensive assessment tool for frailty in everyday clinical practice, (b) a validated measure of cardiovascular risk in IBD, and (c) potential preventive measures against thromboembolism. Vaccination, still frequently underutilized, remains the only efficacious measure against the risk of infections.

## Figures and Tables

**Table 1 cells-12-01722-t001:** Drug–drug interactions with the licensed JAK inhibitors for UC; CYP: Cytochrome P450; OAT3: Organic anion transporter (see [59,60,61,62,63]).

Therapeutic Principle	Metabolism/Elimination	Dose Reduction Needed	Drug-Drug Interactions	Drugs Needing Dose Reduction of JAK Inhibitors (or Need to Be Avoided; A)	JAK Efficacy Weakened
Tofacitinib	liver 70% kidney 30%	severe renal impairment moderate hepatic impairment (should not be used in severe hepatic insufficiency)	strong CYP3A4 inhibitors moderate to strong CYP3A4 inhibitors in combination with CYP2C19 inhibitors strong organic anion transporter (OAT3) inhibitors	ketoconazole, A: Cyclosporin, tacrolimus, A: Grapefruit juice fluconazole	rifampicin
Filgotinib	urines (>80%)	severe renal impairment	not reported	-	-
Upadacitinib	urines (20%)	none	not reported	-	rifampicin

## Data Availability

Not applicable.

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
