# Peer review of "Efficacy and Safety of Biological Therapies and JAK Inhibitors in Older Patients with Inflammatory Bowel Disease"

_cells, 2023, doi:10.3390/cells12131722_

Round 1
Reviewer 1 Report
the problem of older IBD patients-formerly ignored with the misleading idea that they have a mild course of disease-is becoming increasingly important.
this is a short review of the safety of current agents in this special population.
the topic is important and I think it will be interesting for the readers; I liked the way the authors presented and interpreted the data and their conclusions.
Author Response
we the referee for its positive evaluation
Reviewer 2 Report
I think that this article focuses on the challenges and implications of new therapies for inflammatory bowel diseases (IBD), especially in relation to the older patient population. The authors acknowledge that the treatment landscape for IBD has significantly improved and diversified over the last five years, thanks to the introduction of monoclonal antibodies and Janus-Kinase (JAK) inhibitors. These therapies are designed to specifically target different mediators of the immune system, which is important in managing the inflammatory response seen in IBD. The aim of this review, according to the abstract, is to analyze existing literature from pivotal trials and real-world studies to assess the effectiveness and safety of these new treatments in older IBD patients. Special attention is given to the roles of comorbidities and frailty as these factors often coincide in the older population and could have a significant impact on treatment outcomes. The authors note that age itself seems to be of minor importance when compared to these other factors.
This paper is very well-structured, and there is nothing to delete or add.
Author Response
we thank the referee for its/her positive evaluation
Reviewer 3 Report
This is an excellent and comprehensive review and analysis of an important and often neglected area of IBD therapeutics. There are 2 recent relevant studies from McGill that are relevant and should be cited.
1: Hahn GD, Golovics PA, Wetwittayakhlang P, Santa Maria DM, Britto U, Wild GE, Afif W, Bitton A, Bessissow T, Lakatos PL. Safety of Biological Therapies in Elderly Inflammatory Bowel Diseases: A Systematic Review and Meta-Analysis. J Clin Med. 2022 Jul 29;11(15):4422. doi: 10.3390/jcm11154422. PMID: 35956040; PMCID: PMC9369299.
2: Hahn GD, LeBlanc JF, Golovics PA, Wetwittayakhlang P, Qatomah A, Wang A, Boodaghians L, Liu Chen Kiow J, Al Ali M, Wild G, Afif W, Bitton A, Lakatos PL, Bessissow T. Effectiveness, safety, and drug sustainability of biologics inelderly patients with inflammatory bowel disease: A retrospective study. World J Gastroenterol. 2022 Sep 7;28(33):4823-4833. doi: 10.3748/wjg.v28.i33.4823. PMID:36156919; PMCID: PMC9476849.
Author Response
we thank the referee for its/her positive evaluation; the suggested studies are now included